# Correlation of fluorescence microscopy, electron microscopy, and NanoSIMS stable isotope imaging on a single tissue section

Céline Loussert-Fonta [1✉], Gaëlle Toullec[1], Arun Aby Paraecattil[2], Quentin Jeangros[3], Thomas Krueger [1], Stephane Escrig[1] & Anders Meibom [1,4]

Correlative light and electron microscopy allows localization of specific molecules at the ultrastructural level in biological tissue but does not provide information about metabolic turnover or the distribution of labile molecules, such as micronutrients. We present a method to directly correlate (immuno)fluorescent microscopy, (immuno)TEM imaging and Nano-SIMS isotopic mapping of the same tissue section, with nanometer-scale spatial precision. The process involves chemical fixation of the tissue, cryo sectioning, thawing, and air-drying under a thin film of polyvinyl alcohol. It permits to effectively retain labile compounds and strongly increases NanoSIMS sensitivity for $^{13}$C-enrichment. The method is illustrated here with correlated distribution maps of a carbonic anhydrase enzyme isotype, β-tubulin proteins, and $^{13}$C- and $^{15}$N-labeled labile micronutrients (and their anabolic derivates) within the tissue of a reef-building symbiotic coral. This broadly applicable workflow expands the wealth of information that can be obtained from multi-modal, sub-cellular observation of biological tissue.

[1] Laboratory for Biological Geochemistry, School of Architecture, Civil and Environmental Engineering, Ecole Polytechnique Fédérale de Lausanne (EPFL), CH-1015 Lausanne, Switzerland. [2] Greenlight Syntax Sarl, Avenue de Morges 41, 1004 Lausanne, Switzerland. [3] Photovoltaics and Thin-Film Electronics Laboratory, Institute of Microengineering, École Polytechnique Fédérale de Lausanne (EPFL), CH-2002 Neuchâtel, Switzerland. [4] Center for Advanced Surface Analysis, Institute of Earth Sciences, University of Lausanne, CH-1015 Lausanne, Switzerland. ✉email: celine.loussert@epfl.ch

State-of-the-art studies of complex biological processes often combine multiple experimental methods and employ a variety of optical-, electron-, and ion microscopy techniques to image tissues at the single-cell level. These techniques have enabled advances in biology and life science by providing information about the localization and distribution of specific molecules (e.g., fluorescence in situ hybridization (FISH)[1] and immunolabelling[2]), as well as anabolic turnover and cellular exchange processes (e.g. NanoSIMS[3,4]), together with ultrastructural information in even the most complex biological tissues. When these techniques are used in combination, the approach is commonly referred to as correlative microscopy[5].

Although correlative microscopy now features prominently in studies of biological tissue, important limitations still exist with regard to how well different types of imaging information can be correlated. These limitations are primarily due to sample preparation constraints. For example, in correlative light- and electron microscopy (CLEM) most protocols involve two main steps, starting with live or fixed-cell fluorescence imaging, followed by sample preparation for electron microscopy (EM)[6]. Classical EM sample preparation involves chemical fixation, heavy metal staining, dehydration with solvents, resin embedding, and subsequent (ultra) thin sectioning (Fig. 1 left panel). Following this treatment, the soluble compounds in the cell (i.e., cytosolic components) are either thoroughly displayed[7] or completely lost, leading to a shrinkage of the tissue. Altogether it is a different sample compared to the material that was observed previously by fluorescence microscopy. To overcome this problem, a sample preparation method developed by Prof. Tokuyasu[8] is often used in CLEM[9,10], permitting fluorescence and EM to be carried out on the same section[9]. This method involves chemical fixation similar to classical preparation, cryo sectioning, and thawing at room temperature. By avoiding dehydration and resin embedding, the Tokuyasu method minimizes morphological artifacts (in particular tissue shrinkage) and chemical modifications at the molecular level, preserving the (auto-)fluorescence properties of the sample[10] and (to variable degree) its antigenicity, thus permitting a variety of antibody-labeling.

In the last 10–15 years, ultrahigh resolution (ca. 100 nm lateral resolution) quantitative ion microprobe imaging (NanoSIMS), combined with experiments that introduce stable isotopic (e.g., $^{13}C$ and $^{15}N$) and/or elemental labels into a tissue, has made it possible to study anabolic turnover and track-specific molecules with sub-cellular resolution. NanoSIMS imaging has found applications across numerous disciplines within the environmental, biological, and life sciences[3,4,11]. Nevertheless, it is still not possible to correlate information obtained with all three imaging techniques (i.e., fluorescence microscopy, EM, and NanoSIMS) from one-and-the-same section of a biological tissue. Such a capability would represent a major breakthrough in correlative microscopy[12] because it would permit structural, molecular, and anabolic/metabolic information to be directly correlated at the subcellular level. Here we present a methodology, building upon the Tokuyasu[13] method, that enables direct correlation of (immuno)fluorescent microscopy, (immuno)TEM, and NanoSIMS ultra high-resolution stable isotopic mapping of the same biological tissue section (Fig. 1 right panel).

We demonstrate this additional level of correlative microscopy on tissue from a symbiotic coral (here *Stylophora pistillata*, Fig. 2a). Reef-building corals are highly complex organisms that consist of a wide range of tissue and cell types. In these organisms, the ectoderm and endoderm of both oral and aboral layers are separated by a hydrogel-like matrix (mesoglea; black arrowheads in Fig. 2b, c). Many of the endodermal cells host symbiont, photosynthesizing dinoflagellate algae inside symbiosomes[14]. A diverse community of bacteria[15] adds to the complexity of this symbiotic organism, which is referred to as the "coral holobiont". The structural complexity of symbiotic corals thus represents a methodological challenge. At the same time, it conveniently provides the opportunity to demonstrate our sample preparation and correlative microscopy approach on a range of different matrices within a single biological tissue.

## Results

The method presented here builds on the Tokuyasu method with a number of significant improvements that results in compatibility with NanoSIMS stable isotopic imaging, while allowing efficient immunolabeling and preservation of tissue ultrastructure at the TEM level.

**Preserving antigenicity with light chemical fixation**. Some degree of chemical fixation is required to preserve the tissue and cell ultrastructure[16]. One of the commonly used fixatives for EM is a mixture of 2.5% glutaraldehyde and 4% formaldehyde in Sorensen buffer. With this combination, the rapidly penetrating monoaldehyde temporally fixes the specimen until the slower penetrating dialdehyde irreversibly crosslinks proteins.[17] This crosslinking preserves the ultrastructure of the sample but has deleterious effects on immunolabelling, because it interferes with the antigen epitopes (partially or totally). In order to minimize the loss of antigenicity while preserving the capability to obtain high quality ultrastructural imaging by TEM, we performed a series of cryo preparations with increasing concentration of glutaraldehyde (from 0 to 2.5% (vol/vol)) and 4% formaldehyde. In the context of our work, we found that a mix of 0.5% glutaraldehyde and 4% formaldehyde preserves the tissue ultrastructure (TEM-imaging resolution) and optimizes the preservation of tissue antigenicity; this fixation procedure can be adapted to specific biological tissue and antibodies.

**NanoSIMS compatibility**. Following tissue fixation, the classical Tokuyasu method[13] involves cryo protection, cryo sectioning, and air drying. The drying step requires embedding of the wet section in a methyl cellulose uranyl acetate film (MCUA; Fig. 3a), which prevents the section from collapsing and damage to the ultrastructure. However, this film represents an almost impenetrable physical obstacle for NanoSIMS imaging, which requires that the primary ion beam is capable of removing any coating and begin sputtering secondary ions from the sample itself on a short time scale (order of minutes). The MCUA film cannot be removed on a time scale that renders NanoSIMS imaging feasible. Therefore, to benefit from the advantages offered by the Tokuyasu sample preparation method and be able to perform NanoSIMS imaging on the same tissue section, it was necessary to develop a technique to coat and dry a wet cryo (ultra) thin section with an extremely thin film. This film must maintain sample integrity and preserve ultrastructure during air drying and be easily removed during pre-sputtering to enable efficient NanoSIMS imaging. As we show in the following, spin coating with a polyvinyl alcohol (PVA) aqueous solution as embedding medium achieves these objectives and has a number of added advantages.

PVA in aqueous solution has previously been used as an alternative to methylcellulose for cryo section embedding[18]. PVA is highly hydrophilic[19] and thus reduces the air/water surface tension during the drying process. It is also characterized by low viscosity (~4 mPa s) compared to the conventional methylcellulose uranyl acetate solution (~1000 mPa s), thus allowing formation of thin films. The conventional method for the drying step in the Tokuyasu method makes use of a wire loop to hold the cryo section while embedding it in a drop of MCUA. This drop is then gradually sucked up by contact with a filter paper, as the

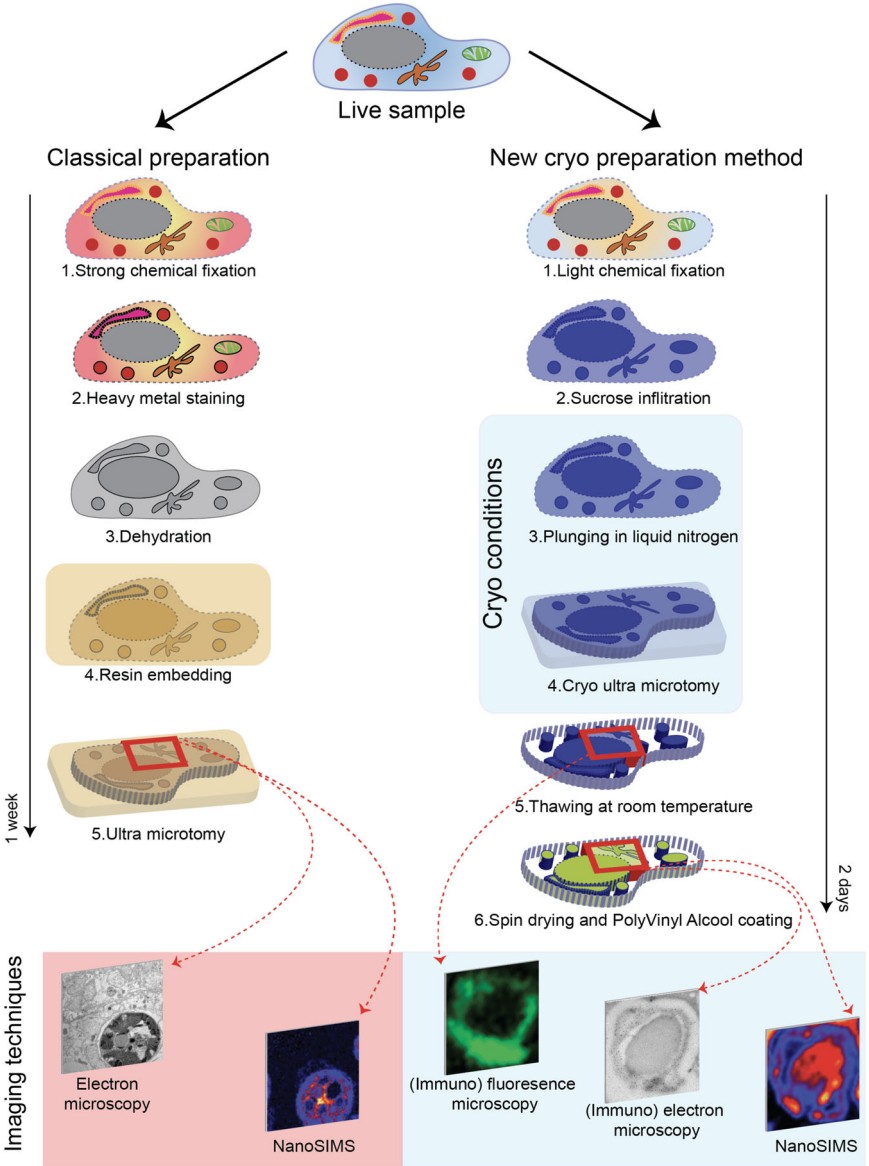

**Fig. 1 Comparison of the classical resin embedding workflow (left panel) and the cryo sectioning methodology introduced here (right panel).** The imaging techniques enabled by each workflow are indicated at the bottom of the figure and are performed on identical ROIs illustrated by the red squares. Regarding cryo preparation, the correlative workflow is performed sequentially, starting with (immuno)fluorescence microscopy on wet cryo section, then followed by (immuno)electron microscopy, and NanoSIMS imaging on the same section that is spin dried and coated with polyvinyl alcool. It has to be noted that the cryo preparation process can be performed in 2 days, versus an entire week for the classical resin-embedding preparation.

wire loop is swept across its surface until no more MCUA solution can be removed from the loop area. This procedure requires substantial dexterity and produces films with a thickness on the order of 75 nm on top of the thin section (Fig. 3a). Introducing a low viscosity PVA aqueous solution as embedding medium yields a thinner film on a wet cryo section when using a spin dryer (cf. Online Methods) (Supplementary Fig. 1). With this approach, which does not require the same level of training and skill as the formation (by hand) of a MCUA film, a film of PVA with a thickness of 15–20 nm can be rapidly and reproducibly formed on top of a wet thin section (Fig. 3b). The ultrastructure of the tissue is comparably maintained by the two drying methods (Fig. 3c, d). Note the stark difference in the width of the gel-like mesoglea in Fig. 3c, d compared to Fig. 2c; dehydration with ethanol during classical sample preparation results in a strong shrinkage of hydrated matrices, which is avoided with our (and the Tokuyasu) method.

**Correlating NanoSIMS isotopic imaging with TEM- and fluorescent microscopy.** In order to correlate (immuno)fluorescence-, (immuno)EM-, and NanoSIMS isotopic imaging, a sample holder suitable for all three imaging modalities is required. Here we used TEM grids as carriers of the cryo thin sections. A wet, thin section supported by a TEM grid is fully compatible with fluorescent microscopy[10]. However, the fragility of TEM grids means that they require a protective support during spin drying. We used a polytetrafluoroethylene (PTFE)-coated rubber septa with a hydrophilic top-surface that maintained the TEM grid in place during spin coating (Supplementary Fig. 1); following spin drying the TEM grid was easily handled for subsequent TEM and NanoSIMS-imaging technics (Fig. 1).

With this workflow, the same cryo thin section (with a thickness of ca. 100 nm) is sequentially imaged in wet conditions by fluorescence microscopy, and then, after drying, by TEM and NanoSIMS. This is demonstrated in Fig. 4, in which—for

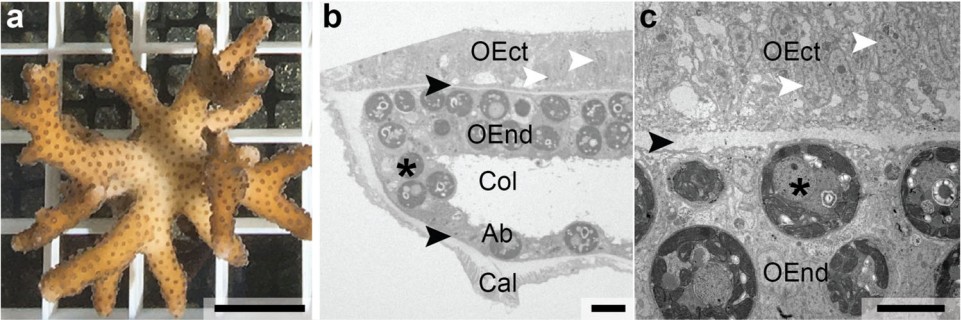

**Fig. 2 The symbiotic coral *Stylophora pistillata* used as model organism in this study. a** A small coral colony. **b** Histological section of a decalcified piece of coenosarc tissue (i.e., between two polyps). The coenosarc is divided into the oral and aboral tissues, subdivided into oral ectoderm, oral endoderm, aboral endoderm, and aboral ectoderm (calicoderm). The oral ectoderm is directly facing seawater and the calicoderm is facing the skeleton. Oral ectoderm/oral endoderm and aboral endoderm/calicoderm are separated by a gel-like mesoglea (black arrows). Many oral endodermal cells host photosynthesizing dinoflagellate algae symbionts (*Symbiodinium;* one marked by an asterisk) surrounded by a symbiosome membrane. **c** The mesoglea interface between the oral ectoderm and oral endoderm is shown in TEM following classical sample preparation (cf. Online Methods). Note the narrow width of the mesoglea. White arrowhead: nucleus; asterisk: dinoflagellate symbiont; black arrowhead: mesoglea; OEct oral ectoderm, OEnd oral endoderm, Col coelenteron, Ab aboral endoderm. Cal calicoblastic ectoderm. Scale bars: **a** 1 cm; **b** 10 μm; and **c**: 5 μm.

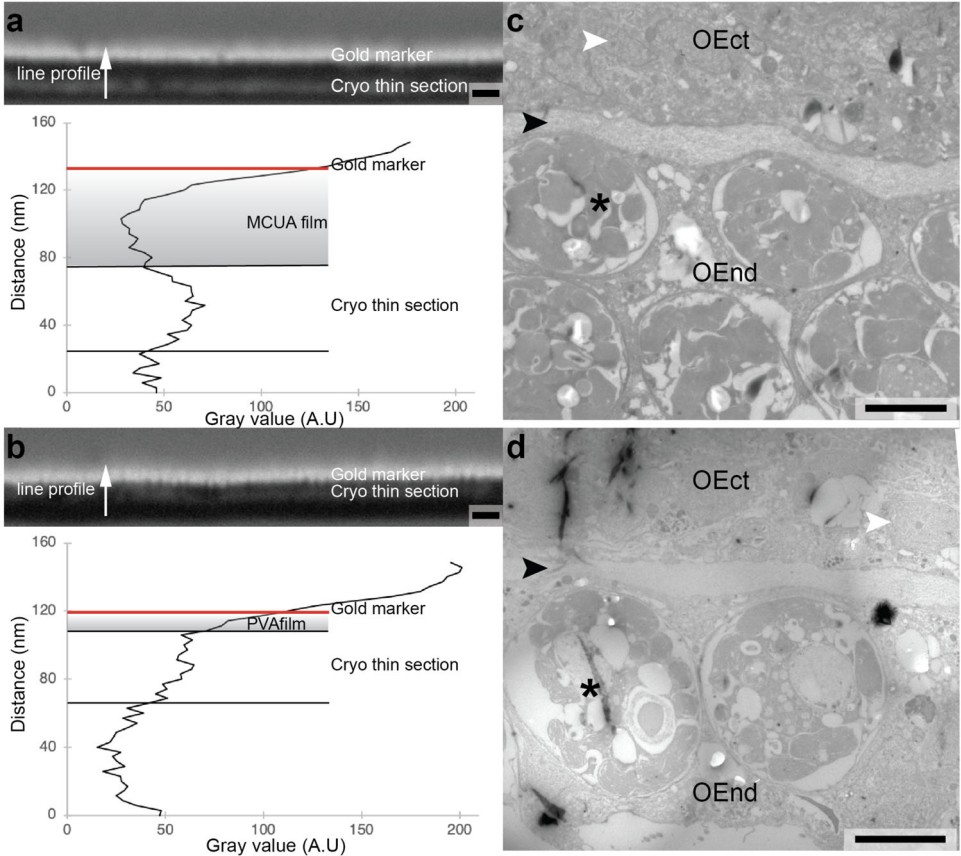

**Fig. 3 Secondary electron scanning electron micrographs of focused ion beam-prepared (FIB) cross-sections of the cryo sections deposited on formvar carbon films (supported on Cu grids) and coated with either MCUA (a) or spin-coated with PVA (b).** Both thin sections were stained with heavy metals to enhance image contrast. To avoid charging and facilitate contrast differentiation with the overlying protective layer of carbon deposited in the FIB section, a 10 nm layer of Au was deposited on top of the MCUA or PVA films. Gray scale intensity profiles taken at the positions indicated by red arrows in **a** and **b** show the difference in thickness of MCUA and PVA films (indicated as shaded areas). **c** and **d** TEM images of the oral ectoderm–endoderm interface in *S. pistillata* obtained with the MCUA layer deposited with normal Tokuyasu sample preparation and with the thin PVA film (this method), respectively. Note the width of the mesoglea (black arrowheads). White arrowheads indicate nuclei; asterisks: symbionts; OEct oral ectoderm, OEnd oral endoderm. Scale bars: **a** and **b**: 100 nm; **c** and **d**: 5 μm.

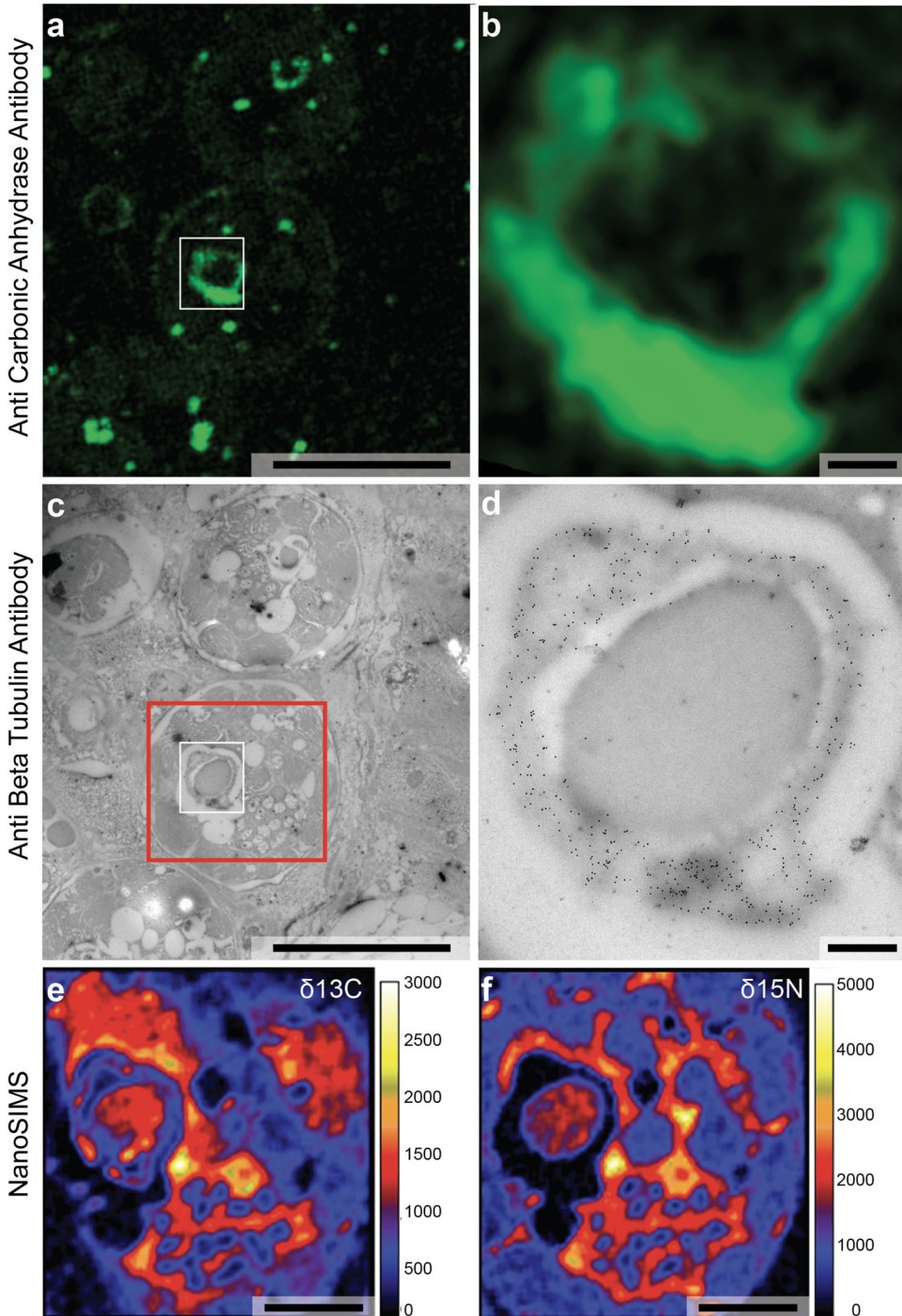

**Fig. 4 The correlative workflow combining immuno-fluorescence microscopy and immuno-electron microscopy with NanoSIMS imaging on the same tissue section of a coral after incubation with H$^{13}$CO$_3^-$ and $^{15}$NO$_3^-$ (and therefore enriched in $^{13}$C and $^{15}$N); cf. Online Methods.** The cryo section (ca. 100 nm thick) was imaged after simultaneous immunolocalization of carbonic anhydrase with anti-Alexa-associated secondary antibodies (**a** and **b**) and β-tubulin observed in TEM by 10 nm gold particles associated to the secondary antibody (**c** and **d**). The fluorescence microscope images (**a** and **b**; zoom in **a**) and the TEM micrographs (**c** and **d**; zoom in **c**) exhibit identical areas of the same thin section. **e** and **f** are NanoSIMS images showing the $^{13}$C and $^{15}$N distributions in the interior of a dinoflagellate symbiont. The area imaged is indicated by a red square in **c**. Scale bars: **a** and **c**: 5 μm; **b** and **d**: 500 nm; **e** and **f**: 2 μm.

illustration purposes—the localization of one isotype of carbonic anhydrase (CA) enzyme is revealed by fluorescence microscopy (Fig. 4a, b), the localization of the β-tubulin proteins by TEM (Fig. 4c, d), and the distribution of $^{13}$C- and $^{15}$N-enrichments by NanoSIMS imaging (Fig. 4e, f) on the exact same area of a thin

section of the symbiotic coral *S. pistillata*, incubated for 6 h in seawater with 2 mM H$^{13}$CO$_3^-$ and 3 μM $^{15}$NO$_3^-$ under natural daylight illumination (cf. Online Methods).

CA is a metallo-enzyme essential to photosynthesis in the symbiont dinoflagellate algae, where it reversibly catalyzes the

conversion of $CO_2$ into carbonic acid, bi-carbonate ions, and protons. The anti-CA2 antibody used for this experiment was raised against a peptide sequence from the human CA2 protein. We matched several proteins from *S. pistillata* and *Cladocopium* (previous nomenclature *Symbiodinium* clade C)[20] by BLASTp. A maximal identity score of 50.2% was obtained for the CA2 isoform X1 (accession number XP 022794253.1) for the host coral, and of 27.2% for the CA14 isoform associated with the dinoflagellate symbionts (accession number OLQ02879.1). With this anti-CA2 antibody we observed CA to be concentrated in the pyrenoid structure and in secondary starch granules inside dinoflagellate cells, and at the vicinity of the symbiosome membranes (Fig. 4a, b); we did not observe distinct localization of CA in the coral host tissue, most likely due to the non-denaturing conditions used for the immuno-labelling. Negative control samples exhibited no fluorescence.

After spin drying and deposition of a ca. 20 nm film of PVA on the thin section, the same region of the section was subsequently imaged by TEM to reveal the location of β-tubulin, using a secondary antibody linked to 10 nm gold particles. The commercial anti-β-tubulin antibody used for this work is known to react with a broad spectrum of β-tubulin isotypes[21], many of which are present in both *S. pistillata* and in *Symbiodiniaceae*. We identified three sites of β-tubulin localization in the coral holobiont: the motile cilia on the outer surface of the host's oral ectoderm (i.e., the coral host; Supplementary Fig. 2A and A') and in the symbionts, inside the pyrenoids, in a shell-like structure embedded in the primary starch sheet and surrounding the central pyrenoid matrix (Fig. 4c, d; Supplementary Fig. 2B), and around starch granules (Supplementary Fig. 2C). We speculate that the structures inside the pyrenoid containing β-tubulin proteins might contain pyrenoid tubules[22]. The co-localization of CA and β-tubulin proteins in a layer surrounding the electron dense pyrenoid matrix is consistent with their presumed role in concentration and delivering of $CO_2$ to the Rubisco-rich center of the pyrenoid, as also described for the marine diatom *Phaeodactylum tricornutum*[23].

Finally, the same thin section was coated with ca. 10 nm Au (standard procedure to avoid surface charging) and transferred to the NanoSIMS ion-microprobe in which the distribution of $^{13}C$- and $^{15}N$-enrichments were mapped in the same region[24]. Following pre-sputtering that rapidly removed the Au coating and the PVA film, the primary $Cs^+$ ion beam was focused to about 100 nm and rastered across the sample surface producing of secondary ions, notably $^{12}C^{12}C^-$, $^{13}C^{12}C^-$, $^{12}C^{15}N^-$, and $^{12}C^{14}N^-$. These ions were extracted, separated from potentially interfering ions in the mass spectrometer, and individually counted in electron-multiplier detectors[3] (cf. Online Methods). This permitted $^{13}C$- and $^{15}N$-enrichments to be quantified through the count-rate ratios $^{13}C^{12}C^-/^{12}C^{12}C^-$ and $^{12}C^{15}N^-/^{12}C^{14}N^-$, respectively. Isotopic enrichments are presented in the δ-notation, i.e., in parts-per-thousand relative to a control sample of normal isotopic composition, prepared and analyzed in an identical manner.

The lateral resolution of a NanoSIMS image is sufficient to allow precise correlation with both the TEM and the fluorescent microscopy images (Fig. 4; Supplementary Fig. 3). NanoSIMS images revealed enrichment levels of $^{13}C$ and $^{15}N$ inside the symbiont nucleus (except in the condensed chromosomes) and cytoplasm (Fig. 4; Supplementary Fig. 3). The chloroplasts and the shell surrounding the pyrenoid matrix (which also contained CA and β-tubulin proteins) were enriched in $^{13}C$, consistent with an active role in C-metabolism of these organelles[25]. The pyrenoid matrix exhibited both $^{13}C$ and $^{15}N$ enrichments (Supplementary Fig. 3). These observations are consistent with

the notion of a high concentration of constantly renewing RubisCO proteins located in this matrix[22,26].

**NanoSIMS imaging—advantages in comparison with conventional sample preparation.** Our sample preparation method creates several important consequences for NanoSIMS imaging. Because our method avoids ethanol dehydration and resin embedding, the chemical composition of the sample (i.e., the matrix) is very different from a resin-embedded section prepared with classical protocols. We observed that the ionization efficiency, and hence the count rate of key ions such as $CN^-$, is higher from the matrix created with our method. Supplementary Fig. 4 shows a qualitative comparison between the $^{12}C^{14}N^-$ count-rates obtained with identical analysis conditions on two comparable coral tissue sections (i.e., similar sample areas, tissue regions, and symbiont density), prepared classically vs. our method (Fig. 1). With the sample matrix generated with our method, the $^{12}C^{14}N^-$ count-rate is about 1.8 times higher than with classical sample preparation, permitting to either obtain better counting statistics with the same analysis time, or reduce the analysis time correspondingly.

Furthermore, in a direct comparison between coral tissue sections prepared classically (i.e., with dehydration and resin embedding) and with our method (i.e., without dehydration and resin embedding) we measured up to about a factor of 10 higher $^{13}C$-enrichments in the latter (Fig. 5). Dehydration (e.g., with ethanol) contributes to a wash-out of labile molecules not immobilized during the chemical fixation step (such as e.g. bicarbonate ions). Moreover, resin embedding introduces a large amount of C with normal isotopic composition and dilutes tissue $^{13}C$-enrichments[3,27]. This resulted in a strong discrepancy between the $^{13}C$-enrichments originally present in a given tissue and those measured with the NanoSIMS on a classically prepared tissue section. With the method introduced here, labile C-rich compounds were not lost to the same degree from the sample and the absence of resin embedding avoided further dilution of the remaining $^{13}C$-enrichments.

The loss and dilution of the $^{13}C$-enrichment with dehydration and resin embedding depended on tissue/organelle structure and density. The biggest difference in measured $^{13}C$-enrichments between tissue prepared classically vs. our method was observed in the gel-like mesoglea and in coral host cells (Fig. 5a, b). In the dinoflagellate algae the effect was less pronounced because their $^{13}C$-enrichments were primarily concentrated in starch grains and lipid droplets[28], which were less affected by dehydration and resin penetration. Note that, because resin contains little nitrogen, its effect on $^{15}N/^{14}N$ ratios was insignificant (Supplementary Fig. 5).

Short of a pure cryo sample preparation chain (i.e., starting with high-pressure freezing and keeping the sample frozen through all subsequent preparation steps) that would preserve all cellular components in place, the sample preparation method presented here (Fig. 1b) preserves labile components in situ to the highest possible degree. These labile components are often essential metabolites and/or precursors for molecules with higher structural order produced in anabolic processes[29]. However, it takes time for anabolic processes (typically minutes to hours), to transfer a measurable (with the NanoSIMS) isotopic enrichment from a labile precursor molecule into structural molecules[30,31]. Therefore, with experiments in which labile components enter the tissue but are not allowed time to be converted by anabolic processes, our preparation method provides the best chance of imaging the distribution of isotopically enriched labile compounds, at least at the tissue

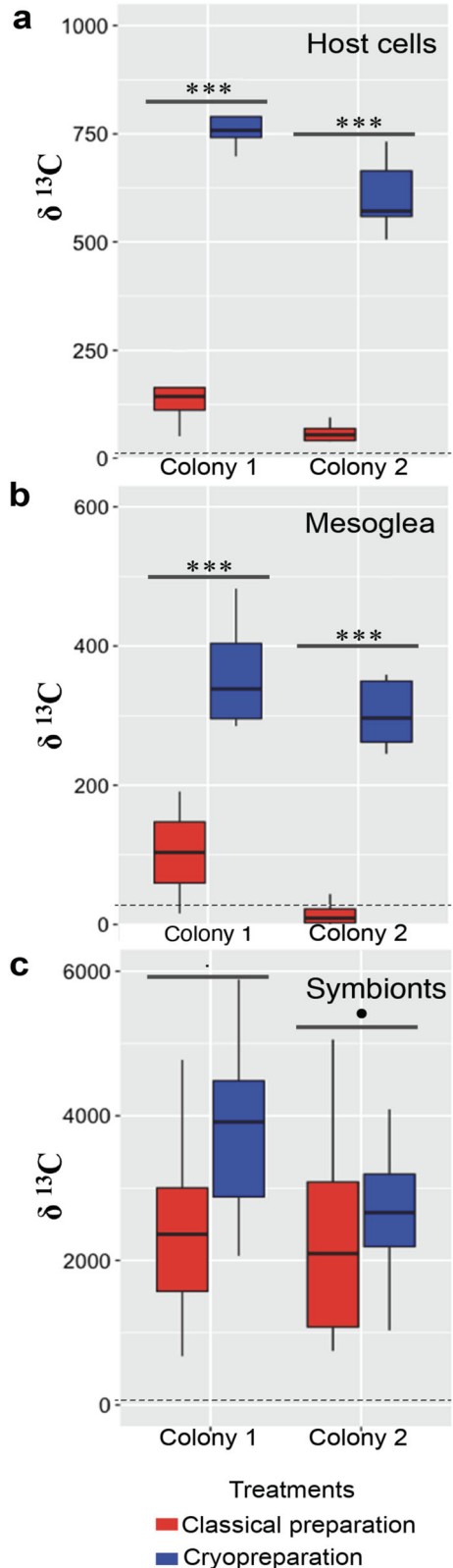

**Fig. 5 Quantitative NanoSIMS analyses of ¹³C-enrichments in two different coral colonies incubated with H¹³CO₃⁻ under identical experimental conditions (cf. Online Methods).** Comparison between classical sample preparation (i.e., with resin embedding) and the cryo sample preparation protocol developed in this study was made for (**a**) coral host cells, (**b**) mesoglea, and (**c**) dinoflagellate symbionts (chemical fixation was 2.5% glutaraldehyde and 4% formaldehyde in both preparations). This sample preparation protocol systematically preserved stronger ¹³C-enrichments in the tissue, by up to a factor of 10, except in the dinoflagellate symbionts, where ¹³C-enrichments were primarily localized in starch granules and lipid droplets, which were less affected by resin dilution. The dashed lines indicate the detection limit for the NanoSIMS, defined here as three standard deviations obtained by analyzing similar regions in unlabeled control tissue ($n = 9$). The statistical differences between the treatments were calculated by one-way ANOVA. The annotation of *p*-value significance level is: "***"[0, 0.001]; "**"(0.001, 0.01]; "*"(0.01, 0.05]; "."(0.5, 0.1]; "" (0.1, 1]; ns non-significant.

## Discussion

We have presented a sample preparation method that allows direct correlation of (immuno)fluorescent microscopy, (immuno) TEM, and NanoSIMS isotopic imaging of a single tissue thin section. We have demonstrated this capability on reef-building symbiotic coral tissue, which represents a methodological challenge because it contains a broad range of biological matrices, from hydro-gel structures to dense starch granules and lipid droplets. Our method creates a number of advantages for NanoSIMS isotopic imaging, notably enhanced ionization efficiency of key secondary ions, and the possibility to map the distribution of isotopically labeled labile components, which are partly retained with our method but lost during classical sample preparation. Our method is applicable to a wide range of biological tissue. This additional level of sophistication in correlated microscopy will find application across the life- and biological sciences, as well as in many branches of the environmental sciences.

## Methods

**Coral maintenance and experimental design.** Specimens of *S. pistillata* were collected from the shallow waters (4–8 m) of the Gulf of Aqaba in Eilat, Israel under permit 2019/42143 of the Israel Nature and Parks Authority. Experiments were conducted in the outdoor Red Sea Simulator system at the Interuniversity Institute (IUI) for Marine Sciences with corals maintained in air-bubbled aquaria (2 L) at ambient water temperature (28 °C) and reduced solar irradiance (ca. 200 μmol m⁻² s⁻¹ at midday under canopies). Three different mother colonies were selected according the similarity of their sizes. For the labeling pulse, two corals were incubated in seawater spiked with 2 mM NaH¹³CO₃ 98 at.% (Sigma-Aldrich, Switzerland) and 3 μM K¹⁵NO₃ 98 at.% (Sigma-Aldrich, Switzerland)[32]. A piece of an unlabeled coral colony was used as reference to assess the natural isotopic ratio in the tissue of interest. This coral was incubated in natural sea water under similar illumination conditions. The experiments were conducted between 10 h 30–16 h 30. The temperature and light illumination were recorded each hour during the 6 h. At the end of the isotopic pulse, fragments of branches at half centimeter from the apical tips were removed, broken into pieces ~1 cm in linear dimension, and immediately transferred into phosphate buffer (0.1 M pH 7.4) with 9% sucrose, containing 4% formaldehyde and glutaraldehyde in different concentrations: 0%, 0.5%, 1%, and 2.5%. Fixation was done at room temperature for 2 h after which the samples were stored in fresh fixative overnight at 4 °C.

**Classical tissue preparation.** After fixation, the samples were decalcified in 0.5 M EDTA solubilized in 0.1 M Sorensen buffer (pH 7.4) containing 0.4% formaldehyde at room temperature for 4 days. Coenosarc tissues were then cut into 1 mm² pieces and post stained in 2% osmium aqueous solution for one hour in the dark. After rinse in water, samples were dehydrated in a series of ethanol concentrations, ranging from 10% to 100% then infiltrated with SPURR resin (Electron Microscopy Sciences, USA) before polymerization at 60 °C for 24 h. Thin sections of 90 nm were cut and mounted onto a Formvar film-coated, carbon-stabilized 100 mesh

level. Conversely, anabolic tissue turnover can be studied by pulse-chase experiments, in which isotopically labeled, labile precursor molecules are permitted to enter the tissue (pulse) and be either metabolized or flushed out of the tissue again (the chase) prior to NanoSIMS imaging.

copper finder grid (Electron Microscopy Sciences, USA) or 200 nm thin sections on silicon wafers (Siltronix, Archamps, France) (Supplementary Fig. 1).

**Cryo preparation**. After fixation and decalcification, coenosarc tissues were casted in 12% gelatin (which does not penetrate into the sample) in 0.1 M Sorensen buffer pH 7.4 and cut into 1 mm³ cubes. Then each sample was cryo protected by infiltration with 2.3 M sucrose (a non-permeating cryoprotectant) in 0.1 M Sorensen buffer pH 7.4 overnight at 4 °C; in this condition the samples can be stored for months. Before cryo thin sectioning, sucrose infiltrated and gelatin-embedded coenosarc pieces were oriented and mounted on aluminum pins under a binocular and immediately plunged into liquid nitrogen (Fig. 1d).

**Cryo sectioning, immunolabelling, and microscopy**. Pins with frozen tissue were mounted in a cryo ultramicrotome (Ultracut UC6/FC6, Leica Microsystems, Austria). Coenosarc blocks were trimmed at −90 °C with a Cryotrim diamond knife (Diatome, Switzerland) and sections of 100 or 200 nm thick were cut at −90 °C with an immuno diamond knife (Diatome, Switzerland). These sections were picked up with a drop of a mixture containing an equal volume of 2% methylcellulose and 2.3 M sucrose[33] to minimize material extraction (compared to pick-up in pure sucrose), warmed up to room temperature, and transferred onto a Formvar film-coated, carbon-stabilized 100 mesh copper finder grid or silicon wafers.

**Histological imaging**. Both resin and cryo sections were mounted onto Superfrost Adhesion microscope slides (ThermoFisher, USA) and stained with a mixture of 1% s toluidine blue and 1% basic fuchsin in water[34] 30 s on 60 °C heat plate. After rinsing in distilled water, the sections were imaged with an upright microscope DM 5500 equipped with CCD camera DFC 3000 B/W (Leica Microsystems, Switzerland) controlled with the Leica software LAS-X (Leica Microsystems, Switzerland).

**Beta-tubulin immuno-gold labeling**. Cryo thin sections collected onto TEM grids were incubated 30 min on gelatin 2% in 0.1 M Sorensen buffer, pH 7.4, at 37 °C to remove the 12% gelatin (used for the embedding of the sample) from the section, then incubated 5 min in PBS buffer containing 1% BSA (Aurion, The Netherlands) as a blocking step to avoid nonspecific labeling. Sections were floated for 1 h on a drop of mouse anti-beta-tubulin antibody (Sigma-Aldrich, Switzerland) diluted at 1/50 in PBS containing 1% BSA. After washing with 0.1% BSA in PBS, the samples were incubated for 1 h with 10 nm colloidal gold-conjugated secondary antibodies, goat anti-mouse (Aurion, The Netherlands) diluted 1:30 in 1% BSA/PBS. The ultrathin cryo sections were then fixed in 1% glutaraldehyde in PBS for 10 min to further stabilize them, followed by eight washes in distilled water, 2 min each. Finally, samples were dried and PVA embedding was done as will be described in the subsection "Drying and embedding of the cryo sections for NanoSIMS analysis".

**CA immuno-fluorescence labeling**. Cryo thin sections were processed as described in the previous subsection with a rabbit anti-CA (reference 100-4157, Rockland, USA) diluted 1/50 as the first antibody and a donkey anti-rabbit linked with Alexa 568 (ThermoFisher, USA) as the secondary antibody. To avoid any autofluorescence, the sections were not post-fixed before imaging. They were then were mounted between a Superfrost Adhesion microscope slides (ThermoFisher, USA) and a 18 × 18 mm Corning® cover glass (Merk, Germany) with PBS containing DAPI[10] as mounting medium. The glass slide/coverslip chambers were sealed with commercial nail polish. Prior to imaging, the samples were kept in the dark, inside a humidity chamber at 4 °C for a maximum of 6 h. Imaging was done using Zeiss LSM 700 inverted confocal microscope (Zeiss, Germany) equipped with Axiocam MRm (B/W) camera (Zeiss, Germany) and operated with the software Zen 2009 (Zeiss, Germany). After imaging, the glass slide/coverslip chambers were disassembled.

When double immunolabelling was performed, the sections were floated on a drop of solution containing both primary antibodies. The secondary antibodies were also mixed in the second incubation step. All other incubation steps were performed as described previously. When immunolabelling was performed on cryo sections mounted on silicon wafers, glass bottom 24-well-imaging plates (MatTek, USA) were used during epifluorescence imaging. The silicon wafers were placed upside down inside the plate chambers onto a 20 µl drop of PBS containing DAPI. Before imaging, the samples were kept in the dark, inside a humidity chamber at 4 °C for a maximum of 6 h. Imaging was done using Zeiss LSM 700 inverted confocal microscope (Zeiss, Germany) equipped with Axiocam MRm (B/W) camera (Zeiss, Germany) and operated with the software Zen 2009 (Zeiss, Germany). After imaging, samples were rinsed in water prior to drying and PVA embedding, as described next.

**Drying and embedding of the cryo sections for NanoSIMS analysis**. After rinsing in Sorensen buffer (pH 7.4) the antigen/antibody bonds were stabilized in 1% glutaraldehyde in PBS for 10 min. After eight washes in distilled water (2 min each), the sections were incubated for 5 min in aqueous solution of 3% PVA (Sigma-Aldrich, Switzerland) followed by spin drying at 66 rotations per second for 45 s to produce a thin film of PVA on the wet section (Supplementary Fig. 1). If silicon wafers or coverslips were used as support for the cryo thin sections during spin drying, no special sample holder was required. When TEM grids were used as support, these had to be protected. A PTFE-coated rubber septa (Agilent, USA) was found to be an ideal support for TEM grids, because the PTFE side is providing a hydrophilic top-surface on which a grid was kept in place during spin drying, following which it was easily removed for subsequent imaging in TEM and NanoSIMS (Supplementary Fig. 1).

**SEM imaging**. Secondary electron SEM images of FIB-prepared cross sections (Fig. 3a, b) were obtained in a Zeiss NVision 40 (Zeiss, Germany). The procedure for cross-sectional imaging involved the deposition of a carbon protection layer, first with the electron beam and then with a gallium beam, on Au-coated layer stacks. Cross-sections were prepared with a gallium ion beam current of 3–1.5 nA (at an acceleration voltage of 30 kV). In-lens secondary electron SEM images were acquired with a beam voltage of 2 kV.

**TEM imaging**. Thin sections were imaged with a Tecnai-12 transmission electron microscope (ThermoFisher, USA) operating at 100 kV with a FEI eagle camera (ThermoFisher, USA) using TIA software (ThermoFisher, USA).

**NanoSIMS imaging**. Thin sections were gold-coated to a thickness of ca. 10 nm prior to mapping of $^{13}$C- and $^{15}$N-enrichments by a NanoSIMS 50 L ion microprobe (Cameca, France) using a 16 keV primary Cs$^+$ ion beam. Following pre-sputtering, the primary beam was focused to a spot-size of around 100 nm and rastered across the sample surface in selected regions (guided by fluorescence and TEM imaging) with areas of 40 × 40 or 25 × 25 µm², 256 × 256 pixels, and a pixel dwell-time of 5000 µs. The secondary ions $^{12}C_2^-$, $^{13}C^{12}C^-$, $^{12}C^{14}N^-$, and $^{12}C^{15}N^-$ were separated from potential interferences at a mass resolution of around 9000 (Cameca definition) and counted individually and simultaneously in electron multipliers. Each finished image consisted of five rastered images (except for the images displayed in Fig. 4, which consist of 30 layers) added together following drift correction using the L'Image© software developed by Dr. Larry Nittler.

**Statistics and reproducibility**. The NanoSIMS data presented in this work represent 171 dinoflagellate symbionts (with an apparent diameter larger than 5 µm to ensure a close to equatorial cut through the symbiont) from two colonies, and 26 image frames of 40 × 40 µm for host tissue and mesoglea. To assess the effect of sample preparation on NanoSIMS measurements, we analyzed the data by one-way ANOVA.

**Reporting summary**. Further information on research design is available in the Nature Research Reporting Summary linked to this article.

## Data availability

The datasets generated during and/or analyzed during the current study are available from the corresponding author on reasonable request. Source data are avaialable as Supplementary Data 1.

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

## Acknowledgements

This work was funded by Swiss National Science Foundation Grant 200021_179092 to A. M. We thank Prof. Maoz Fine for access to his Red Sea Simulator aquarium system at the InterUniversity Institute for Marine Science in Eilat and for insightful technical discussions. Drs. Prof. Isabelle Domart-Coulon, Savary Romain, and Nils Rädecker are thanked for constructive criticism of earlier versions of this manuscript.

## Author contributions

C.L.-F., G.T., T.K., and A.M. designed the experiments. C.L.-F. and G.T. conducted the experiments. A.A.P. provided support for film coating. Q.J. performed FIB-SEM imaging. S.E. provided support for NanoSIMS anaysis. C.L.-F. analyzed the samples and performed data analyis. C.L.-F. and A.M. produced the manuscript.

## Competing interests

The authors declare no competing interests.
