## [Peer Review File · Communications Biology]

REVIEWERS' COMMENTS:

Reviewer #1 (Remarks to the Author):

In this article, the authors propose a novel correlative imaging method combining fluorescence microscopy, electron microscopy and stable isotope imaging. The authors optimize chemical fixation to preserve tissue antigenicity, modify the Tokoyasu method by introducing polyvinyl alcohol aqueous solution as an embedding medium that maintains sample integrity and preserves ultrastructure, and adapt handling of transmission electron microscopy (TEM) grids to allow imaging using the three techniques mentioned above. The method was validated through the tissue analysis of the symbiotic coral *Stylophora pistillata* upon labeling of the carbonic anhydrase (fluorescent imaging), of beta-tubulin (TEM imaging) and imaging of ^{13}C and ^{15}N (nanoSIMS). The ionization efficiency and ion rate count is more efficient in the new proposed method than in the classical counterpart method. This method therefore allows for the first time to correlate, using the same sample, three major imaging techniques with preserved antigenicity and enhanced detection of isotopically labeled labile compounds, providing a major advancement for ultrastructural studies.

Reviewer #2 (Remarks to the Author):

Quantitative materials analysis is an important emerging frontier in life science microscopy. A powerful example is the correlation between ultrastructural electron microscopy and fluorescence imaging, possibly at resolutions beyond optical diffraction. This particular combination has largely captured the label of "correlative microscopy", but it is only the most obvious. Soft X-ray tomography and scanning transmission electron microscopy can provide quantitative density measurements, while localized spectroscopies including EELS, EDS, and Raman can detect the presence of particular elements or chemical bonds. Mass spectrometry goes a step further to detect isotopic ratios. This opens exciting possibilities for pulse-chase experiments, as employed for example by Ellisman and Hetzer to study protein turnover rates.

Correlative analysis confronts two important challenges. The first is that specimen preparation must be compatible with more than one imaging modality, and the second, not entirely independent, is that the specimen preparation should not alter the material composition to be analyzed. Cryogenic preparation by vitrification is the gold standard, but it is not always applicable and constrains the subsequent analysis quite strictly. Too often, unfortunately, the problem is just ignored, especially in conventional electron microscopy based on dehydration and plastic embedding. Soluble components in the original specimen are washed away, possibly causing dissolution of remaining structures, and analysis continues with what remains.

The manuscript by Loussert-Fonta et al describes in detail a newly developed extension of the Tokoyasu protocol for preservation compatible with fluorescence, TEM, and nanoSIMS imaging. It is based on spin coating of an ultrathin, low viscosity polyvinyl alcohol film instead of the more conventional methylcellulose overlay. In addition, the fixative concentration was calibrated to maintain antigenicity for immuno-fluorescence labelling. The contribution of the work is not so much in inventing new tricks as in carrying the integrated workflow all the way through to a productive goal. As a reader I appreciated very much the focus on avoiding loss of labile components and the comparison with cryopreservation. Preservation of carbon isotopes is especially incompatible with conventional plastic embedding. In summary, the new approach enables nanoSIMS analysis of tissue sections in direct correlation with fluorescence and transmission electron microscopy, and I fully expect that parts of the protocol will be adopted more broadly. I am happy to recommend publication in *Communications Biology*.

Reviewer #3 (Remarks to the Author):

The authors describe a workflow for preparing biological tissues for correlative imaging with

fluorescent microscopy, transmission electron microscopy, and nanoscale secondary ion mass spectrometry.

The novelty in this study is that the proposed sample preparation method enables correlative imaging to be performed on the same area of a tissue. This approach is demonstrated on coral. The sample preparation procedure described is a modified version of the widely applied Tokuyasu method where spin coating is used to enable deposition of thin films of poly vinyl alcohol (PVA) onto sections of coral enabling nanoSIMS imaging. The authors also provide evidence that the proposed protocol improves ionization yields of certain elements and stable isotopes in NanoSIMS when compared to classical sample preparation methods.

Overall, this paper was extremely well written. Methods are described in detail, which is necessary when proposing improvements to current sample preparation protocols. I believe that this paper will be of great interest to the readers of Nature Communication Biology and the wider scientific community. To The authors should be commended for their work. I can recommend this paper for publication after some minor issues are addressed.

Minor comments and suggestions

1. Line 55-56. Fixation protocols may also redistribute compounds, including proteins, lipids, and metabolites as established by Huebinger et al. 2018 (<https://doi.org/10.1038/s41598-018-36112-w>). This should be stated as using stable isotope probing with NanoSIMS to quantify isotope enrichment can be affected by fixation redistribution effects introducing measurement error.
2. Line 70. Reference statement of anabolic turnover and metabolic tracking.
3. Line 72. Move period to after parentheses to match formatting for rest of manuscript.
4. Line 110. Can the authors add any relative intensity information related to the carbon anhydrase reporter in the fixation effect study? An additional figure or data set is not required, however some numbers to back this up would be a nice addition.
5. Line 133. Please add reference to hydrophobicity and viscosity values.
6. Line 135. Adding reasoning behind why the authors did not attempt to spin coat methyl cellulose would be useful. If the viscosity of cellulose is the limitation behind spin coating thin films this needs to be referenced. Lua et al. 2007 (<https://doi.org/10.1021/la0629680>) has shown that spin coating methyl cellulose produces thick films.
7. Line 141. What methodology did the authors follow for determining the interface between the section/MCUA film, and MCUA film/gold marker? Can the authors please include this either in the methods section, or in the corresponding caption for figure 3?
8. Line 146. Usually film thickness measurements include how many samples taken, and thickness standard deviation. As this is a methodology paper, the authors should add sample size, and average film thickness with deviation values.
9. Line 155. "A wet, thin".
10. Line 160. This reads as though the samples were removed from the TEM grids, then imaged with TEM. Is this correct?
11. Line 162-163. Reword "This approach enabled to image the same", grammatically incorrect.
12. Line 168. Change "was incubated" to "following incubation", and restructure sentence.
13. Line 206. How were the interfering ions separated? Is separation based purely on flight time in the TOF or was spectral processing performed? Please add details.
14. Line 226. Change "with a resin" to "to a resin".
15. Line 241. "resin embedding introduces massive amounts". Please restructure sentence to remove "massive".
16. Line 251. Figure 5C is not referred to in the text, please add.
17. Line 494. Figure 5 caption needs to include the unit areas where the ion count ratios were derived.

Dear Reviewers,

We are pleased to read that our manuscript raised enthusiasm among you. Also, we are convinced that this new method will find applications across life-, biological, and environmental sciences, thanks to the broad audience of *Communication Biology*.

We modified the text according your comments and suggestions, as detailed below.

On behalf of all co-authors – sincerely yours,

Dr. Loussert-Fonta Céline

Replies to questions/comments made by Reviewer 3:

Comment 1. Fixation protocols may also redistribute compounds, including proteins, lipids, and metabolites as established by Huebinger et al. 2018 (<https://doi.org/10.1038/s41598-018-36112-w>). This should be stated as using stable isotope probing with NanoSIMS to quantify isotope enrichment can be affected by fixation redistribution effects introducing measurement error.

Answer: Reference added line 56 in the text.

Comment 2. Reference statement of anabolic turnover and metabolic tracking.

Answer: The references appear naturally after the next sentence.

Comment 3. Move period to after parentheses to match formatting for rest of manuscript.

Answer: Modified in the text

Comment 4. Can the authors add any relative intensity information related to the carbon anhydrase reporter in the fixation effect study? An additional figure or data set is not required, however some numbers to back this up would be a nice addition.

Answer: The specificity of each antibody at each fixation condition was determined by transmission electron microscopy according to the guidelines proposed by Griffiths et al. (DOI 10.1007/s00418-014-1263-5).

Comment 5. Please add reference to hydrophobicity and viscosity values.

Answer: We added a reference for the hydrophilic parameter (Yang) and 4 mPa per s versus 1000 mPa s for the viscosity parameters as indicated by the manufacturers.

Comment 6. Adding reasoning behind why the authors did not attempt to spin coat methyl cellulose would be useful. If the viscosity of cellulose is the limitation behind spin coating thin films this needs to be referenced. Lua et al. 2007 (<https://doi.org/10.1021/la0629680>) has shown that spin coating methyl cellulose produces thick films.

Answer. We tried to spin coat the methyl cellulose but due to its viscosity, we never succeed in obtaining a thinner film than the one achieved by hand with a filter paper.

Comment 7. What methodology did the authors follow for determining the interface between the section/MCUA film, and MCUA film/gold marker? Can the authors please include this either in the methods section, or in the corresponding caption for figure 3?

Answer: We explain in the figure caption that the grey values are used to separate the layers.

Comment 8. Usually film thickness measurements include how many samples taken, and thickness standard deviation. As this is a methodology paper, the authors should add sample size, and average film thickness with deviation values.

Answer: We investigated 2 TEM grids per condition by FIB-SEM microscope. Due to the costs associated with this instrument, it was not possible to image more samples. We are certain that the thickness reported is consistently reproduced because our coated thin sections behave identically in the NanoSIMS.

Comment 9. "A wet, thin".

Answer: Modified in the text

Comment 10. This reads as though the samples were removed from the TEM grids, then imaged with TEM. Is this correct?

Answer: Modified in the text

Comment 11. Reword “This approach enabled to image the same”, grammatically incorrect.

Answer: Modified in the text

Comment 12. Change “was incubated” to “following incubation”, and restructure sentence.

Answer: Modified in the text

Comment 13. How were the interfering ions separated? Is separation based purely on flight time in the TOF or was spectral processing performed? Please add details.

Answer: The details can be found in the NanoSIMS review papers cited in the text.

Comment 14. Change “with a resin” to “to a resin”.

Answer: Modified in the text

Comment 15. “resin embedding introduces massive amounts”. Please restructure sentence to remove “massive”.

Answer: Modified in the text

Comment 16. Figure 5C is not referred to in the text, please add.

Answer: We discuss Fig 5, excluding 5c.

Comment 17. Figure 5 caption needs to include the unit areas where the ion count ratios were derived.

Answer: This is a misunderstanding. The measured $^{13}\text{C}/^{12}\text{C}$ ratios are average numbers for the regions of interest selected. This is independent of the area and there is no area ‘normalization’ involved. Our manuscript contains references to technical NanoSIMS papers explaining this in more detail.